# Cloning and Characterization of *Drosophila melanogaster* Juvenile Hormone Epoxide Hydrolases (JHEH) and Their Promoters

**DOI:** 10.3390/biom12070991

**Published:** 2022-07-16

**Authors:** Dov Borovsky, Hilde Breyssens, Esther Buytaert, Tom Peeters, Carole Laroye, Karolien Stoffels, Pierre Rougé

**Affiliations:** 1Department of Biochemistry and Molecular Genetics, University of Colorado Anschutz Medical Campus, Aurora, CO 80045, USA; 2Zoological Institute, KU Leuven, 3000 Leuven, Belgium; hilde.breyssens@gmail.com (H.B.); esther.buytaert@kuleuven.be (E.B.); tom.peeters@ehb.be (T.P.); carole_laroye@hotmail.com (C.L.); stoffelskarolien@yahoo.com (K.S.); 3Open BioLab Brussels, Erasmushogeschool Brussels, 1210 Brussels, Belgium; 4Faculte des Sciences Pharmaceutiques, 31400 Tolouse, France; pierre.rouge.perso@gmail.com

**Keywords:** drosophila, juvenile hormone epoxide hydrolases promoters, transcription factors, sequencing, 3D modeling, tissue culture, embryo transformation, RNAi

## Abstract

Juvenile hormone epoxide hydrolase (JHEH) plays an important role in the metabolism of JH III in insects. To study the control of JHEH in female *Drosophila melanogaster*, JHEH 1, 2 and 3 cDNAs were cloned and sequenced. Northern blot analyses showed that the three transcripts are expressed in the head thorax, the gut, the ovaries and the fat body of females. Molecular modeling shows that the enzyme is a homodimer that binds juvenile hormone III acid (JH IIIA) at the catalytic groove better than JH III. Analyses of the three JHEH promoters and expressing short promoter sequences behind a reporter gene (*lac*Z) in *D. melanogaster* cell culture identified a JHEH 3 promoter sequence (626 bp) that is 10- and 25-fold more active than the most active promoter sequences of JHEH 2 and JHEH 1, respectively. A transcription factor (TF) Sp1 that is involved in the activation of JHEH 3 promoter sequence was identified. Knocking down Sp1 using dsRNA inhibited the transcriptional activity of this promoter in transfected *D. melanogaster* cells and JH III and 20HE downregulated the JHEH 3 promoter. On the other hand, JH IIIA and farnesoic acid did not affect the promoter, indicating that JH IIIA is JHEH’s preferred substrate. A transgenic *D. melanogaster* expressing a highly activated JHEH 3 promoter behind a *lac*Z reporter gene showed promoter transcriptional activity in many *D. melanogaster* tissues.

## 1. Introduction

Juvenile hormone (JH) in insects plays a pivotal role in larval metamorphosis [1], regulation of pheromone biosynthesis and vitellogenesis [2]. JH biosynthesis in insects is regulated by neuropeptides (allatotropins and allatostatins) secreted from the brain [3] and by JH esterase (JHE; EC 3.1.1.1) and JH epoxide hydrolase (JHEH; EC 3.3.2.3) [4,5,6]. JHE is a carboxylesterase and hydrolyzed the methyl ester group of JH, converting the enzyme to JH acid (JHA) that can be converted back to JH by JHA methyl transferase (JHAMT) [7,8,9], whereas the metabolism of JHA into JHA diol (JHAD) is irreversible [10,11,12]. Both JHA and JHAD are inactive, and this degradative pathway was reported for *Drosophila melanogaster*, *Manduca sexta*, *Culex quinquefasciatus* and *Aedes aegypti* [6,13,14,15]. The crystal structure of JHEH from *Bombyx mori* has been described showing a homodimeric conformation of JHEH [16]. The *jheh* gene was cloned from *M. sexta*, *Bombyx mori* and *Apis mellifera* [17,18,19]. The expressed *jheh* in the honeybee did not show activity against JH III and probably does not participate in the JH III degradation pathway. A cDNA clone from *D. melanogaster* third instar larva expressing JHEH showed no activity against JH III and was proposed to be involved in xenobiotic biotransformation but not in JH metabolism [20]. Borovsky et al. [12] showed that female *Ae. aegypti* that was treated with [12-^3^H]-(10R)-JH III preferentially metabolized the radioactively labeled JH III into JH IIIAD and JH IIIA at a ratio of acid diol/acid/diol of 17/4/1, showing that JHEH prefers to metabolize JH IIIA and JH III is metabolized first to JH IIIA by JHE and not to JH IIID. These authors also showed that female mosquitoes treated with [12-^3^H]JH IIIA metabolized JH IIIA 17-fold faster into JH IIIAD than they metabolized JH III into JH IIID. To find out how *jheh* is regulated in *D. melanogaster*, it is necessary to study both the *jheh* promoter region that controls *jheh* and its transcription factors (TFs) that regulate the transcription of *jheh* in response to physiological and external stimuli [21,22]. Most eukaryotic TFs have a DNA-binding domain (DBD) and recent studies showed that TFs promote regulatory elements such as enhancers and promoters form looped structures to facilitate transcription regulation [23]. 

To study the function of promoters in insects, several techniques have been developed in which potential promoters are linked to reporter genes such as Luciferase (*luc*) or green fluorescent protein (*gfp*) to study their activity in insect cell lines such as *D. melanogaster* (D.Mel2). Using this approach, a novel 121 promoter from anhydrobiotic midge *Polypedilum vanderplanki* was identified, which was expressed in various insect cell lines [24]. In *Spodoptera frugiperda,* heat shock proteins promoters have been analyzed using Sf9 cells and various heat shock (*SfHsp*) genes that were cloned into a plasmid with a *luc* reporter. Using this technique, a strong heat shock promoter that drove the luciferase gene was found [25], and RNA polymerase-II-dependent promoters were isolated in order to facilitate more efficient protein expression in Sf21 and Hi5 cells [26]. The role of promoters in insect immunity was reported for *D. melanogaster*, in which the diptericin promoter plays an important role in the insect’s immunity [27,28]. To characterize the *Drosophila* core promoter, Qi et al. [29] used synthetic promoters to facilitate large-scale analysis in S2 cell.

To find out how *jheh* is controlled in *D. melanogaster*, we cloned the cDNAs of *jheh* 1, 2 and 3 and followed their expressions using Northern blot analyses of the ovary, gut and head thorax of female *D. melanogaster*. Molecular 3D modeling of the three JHEHs and docking of JH IIIA and JH III into the catalytic groove of JH III allowed us to follow their binding specificities to the catalytic groove of JHEH 3. To find out the JHEH active promoter region, *jheh* 1, 2 and 3 promoters were cloned behind a *lac*Z reporter gene and analyzed in *D. melanogaster* D.Mel2 cells. The effects of JH III, JH IIIA, 20Hydroxyecdysone (20HE) and farnesoic acid on the *jheh* 3 promoter were studied. This report is the first to show how JH III and TF Sp1 regulate the *jheh* 3 promoter of *D. melanogaster*. 

## 2. Materials and Methods

### 2.1. Insects, Cells, Chemicals and Incubation Conditions

*D. melanogaster* were reared at 21 °C on a diet of instant *Drosophila* medium containing blue color (Carolina Biological Supply Company, North Carolina USA). *Drosophila melanogaster* D.Mel2 cells (2 × 10^5^ cells/mL) were grown in serum-free medium (SFM) containing 9% Glutamine (1.5 mL) in 24-well sterile plates (Invitrogen, Waltham MA USA) at 27 °C following the manufacturer’s guidelines. Cells were stored in liquid N_2_ and were split no more than 10–15 times. The cells were checked often for viability with trypan blue, as recommended by the manufacturer. To transfect *D. melanogaster* cells with different promoter sequences, Cellfectin (10 μL) was mixed with SFM (500 μL) containing Glutamine (9%) (Invitrogen, Waltham, MA USA) and plasmid pCaSpeR-AUG-βgal (2 μg) carrying promoter’s sequence to be tested, and the mixture was incubated with D.Mel2 cells (2 × 10^5^ cells) for 3 h. After incubation, the medium was removed, and fresh SFM containing 9% Glutamine was added to each well and the cells were incubated for 72 h at 27 °C before testing for β-galactosidase activity. Transfected D.Mel2 cells were lysed in 0.25 M Tris-HCl pH 8.0 buffer (Invitrogen), and the enzymatic activity of β-galactosidase was followed by measuring the absorbance at 420 nm. The absorbance reading was converted into β-galactosidase activity expressed in milliunits (mU), equivalent to 1 nmole of β-D-galactose hydrolyzed per min, using a purified β-galactosidase with known activity (Sigma, St Louis, MO, USA), and a calibration curve was constructed following the manufacturer’s guidelines. pCaSpeR-AUG-βgal [30] was obtained from the Drosophila Genomics Resource Center Indiana University (Bloomington, IN, USA) and the *D. melanogaster* λ- ZAP II cDNA library of 2-week-old males and females was obtained from Stratagene (La Jolla, CA, USA). Farnesoic acid and [12-^3^H](10R) JH III were provided by Professor G. Prestwich (University of Utah). JH III acid (JH IIIA) and [12-^3^H]JH IIIA were synthesized by hydrolyzing the methyl ester of JH III, converting it into JH IIIA by incubating JH III with 0.5 N NaOH in ethanol for 24 h at room temperature. The JH IIIA was purified by reversed phase C_18_ HPLC and stored in hexane [9,31]. Next, 20-Hydroxy ecdysone (20HE) was obtained from Sigma and its mimic (RH5992) was provided by Professor G. Smagghe (university of Ghent, Ghent, Belgium). TRIzol for RNA extraction from female *D. melanogaster* and D.Mel2 cells was obtained from GIBCO BRL (Gaithersburg, MA, USA), and o-nitrophenyl,β-D galactopyranoside (ONPG) was obtained from Sigma (St. Louis, MO, USA). dsRNA was synthesized using a HiScribe RNAi transcription kit (New England BioLabs, Beverly, MA, USA). A dsDNA was cloned into pLitmus 28i and the dsDNA was converted into dsRNA using T7 and PCR. The integrity of the dsRNA was determined by agarose gel electrophoresis, following the manufacturer’s guidelines.

#### Injections into Embryos

*D. melanogaster* w^1118^/yw (white eye) embryos were injected with pCaSpeR-AUG-βgal carrying JHEH 3 promoter sequence (627 bp) by BestGene (Chino Hills, CA) using p-element transformation. The surviving G_0_ adults were crossed with w^1118^/yw and the G_1_ adults were crossed again with w^1118^/yw, and the resultant G_2_ transformants were balanced. Thirty-two transformed *D. melanogaster* males and females were then assayed for β-galactosidase activity by lightly anesthetizing them with ether. The flies were added to a 96-well microtiter plate containing 50 mM sodium phosphate buffer pH 8.0, 2 mM potassium ferrocyamide, 0.3% X-Gal and 15% Ficoll (100 μL) in each well. *D. melanogaster* abdomens were poked with a finely drawn glass capillary and the staining solution was allowed to penetrate and absorb inside each fly overnight at room temperature in the dark. After incubation, transformed flies were observed under a Nikon-dissecting microscope for areas that were stained blue. Controls that were not transformed were treated similarly.

### 2.2. RNA Extraction and Purification

Total RNA was extracted from *D. melanogaster* tissues and adults (150 per group) with TRIzol reagent (Invitrogen, Carlsbad, CA, USA) according to the manufacturer’s instructions.

### 2.3. Cloning and Sequencing of JHEH 1, 2 and 3 cDNAs

Using the *D. melanogaster* genome (accession number AE003798) and the published sequences of JHEH for *Trichoplusia ni* and *Manduca sexta* [32,33] homologous sequences in conserved regions were used to synthesize primers for PCR amplifications of *D. melanogaster* λ Zap II cDNA library (diluted 1:10) (Stratagene). The PCR protocol was as follows: denaturation for 3 min at 95 °C (1 cycle), annealing for 4 min at 48 °C, extension for 40 min at 60 °C (1 cycle each), denaturation at 95 °C for 30 s, annealing for 30 s at 48 °C and extension for 2 min at 60 °C (40 cycles), with a final extension for 15 min at 60 °C. Following PCR, the dsDNA was separated by gel electrophoresis on 2% agarose gel in tris–acetate–EDTA (TAE) buffer (pH 7.8) containing ethidium bromide at 100 v for 60 min. Amplified DNA bands were visualized under UV light, cut from the gel, eluted with a QIAquick gel extraction kit (Qiagen, Germantown, MD, USA) and cloned into TOPOpCR2.1 according to the manufacturer’s instructions (Invitrogen, Carlsbad, CA, USA). INVαF’ *E. coli* cells were transformed, and plasmids were purified with a QIAprep Spin Miniprep Kit (Qiagen), sequenced using a BigDye Terminator v3.1 Cycle Sequencing Kit (Applied Biosystems, Waltham, MA, USA) and analyzed at the University of Florida DNA sequencing core (https://biotech.ufl.edu/gene-expression-genotyping (accessed on 14 June 2022)). To amplify the 3′ and 5′ ends of *D. melanogaster* JHEHs, RT-PCR reactions (20 μL) containing 4 μL 25 mM MgCl_2_, 2 μL 10× PCR buffer (Applied Biosystems, Foster City, CA), 6 μL sterile distilled water, 4 μL dNTP (10 mM each of dATP, dTTP, dCTP and dGTP), 1 μL RNase inhibitor (20 U) and 1 μL MMLV reverse transcriptase (50 U) were prepared, containing 1 μL reverse primer (15 μM) and *D. melanogaster* total RNA (1 μg). Reverse transcription (RT) was performed in a thermal cycler (Applied Biosystems) at 24 °C for 10 min, followed by 42 °C for 60 min, 52 °C for 30 min, 99 °C for 5 min and 5 °C for 5 min. After RT, 3 μL 10× Buffer, 25.5 μL sterile distilled water, 2.5 U AmpliTaq DNA polymerase (Applied Biosystems, Waltham, MA, USA) and 15 μM of a forward primer were added to the reaction mixture and PCR was carried out, as described above.

#### 2.3.1. Sequencing of JHEH 1

JHEH 1 cDNA was amplified and sequenced using forward and reverse primers (Appendix A), generating overlapping amplicons of 432, 515, 293, 426, 315, 452 and 374 nt that covered the entire JHEH cDNA (Figure 1a). The 3′ and 5′ ends were amplified using rapid amplification ends (RACE) [34] and the entire cDNA (1470 nt) of JHEH 1 sequence was deposited in the GenBank (accession number AF517545). 

#### 2.3.2. Sequencing of JHEH 2

JHEH 2 cDNA was amplified and sequenced using forward and reverse primers (Appendix A) generating overlapping amplicons of 432, 404, 515, 315, 292, 492, 413 and 386 nt covering the entire JHEH 2 cDNA (Figure 1b). The 3′ and 5′ ends were amplified using RACE [34] and the entire cDNA (1543 nt) of the JHEH 2 sequence was deposited in the GenBank (accession number AF517546).

#### 2.3.3. Sequencing of JHEH 3

JHEH 3 cDNA was amplified and sequenced using forward and reverse primers (Appendix A) generating overlapping amplicons of 432, 515, 425, 308, 442, 345, 322 and 298 nt that covered the entire JHEH 3 cDNA (Figure 1c). The 3′ and 5′ ends were amplified using RACE [34] and the entire cDNA (1441 nt) of JHEH 3 sequence was deposited in the GenBank (accession number AF517547).

### 2.4. Identification of JHEH 1, 2 and 3 Promoter Regions and Introns

The *D. melanogaster* genomic scaffold (1420000133386047 Section 8, GenBank accession number AE003798.1) was analyzed using Lasergene Genomic Suite software (DNASTAR) to determine the location of introns, exons and promoter regions of JHEH 1, 2 and 3 (Figure 2).

### 2.5. Northern Blot Analysis

#### 2.5.1. JHEH 1, 2, 3

An Ambion Northern Max kit (Ambion, Foster City, CA, USA) was used for the Northern blot analysis. Total RNA was extracted in Trizol from female *D. melanogaster* isolated guts, fat bodies ovaries and intact females (150 per group) 3 days after emergence. RNA (15 μg/lane) samples that were extracted were separated on denaturing 1.0% formaldehyde agarose gel at 100 V for 1.5 h [35], transferred to Hybond-N^+^ nylon membrane and hybridized with [^32^P]-labeled probes for JHEH 1, 2 and 3 (Appendix A). Hybridized membranes were exposed to X-ray film for 48 h at −80 °C and then developed [35,36] (Figure 3a–c). Radioactively labeled probes were stripped from the membrane in a boiling solution of 0.1% SDS, and the membrane was scanned with a Geiger counter to confirm that the probes were completely removed. The stripped membranes were then hybridized with *D. melanogaster* actin transcript probe (355 bp) (accession number BT050557.1) to show equal transfer to the membrane in all lanes (Figure 3a–c). The actin probe cDNA was prepared by PCR using *D. melanogaster* λ-Zap II cDNA library (Stratagene) with primer pair DB657 (forward) 5′ CAGGTGATCACCATTGGCAACGGAGCG 3′ (*t_m_* 62 °C) and DB658 (reverse) 5′ CCTGCTTCGAGATCCACATCTGCTG 3′ (*t_m_* 63 °C). The cDNA actin probe was labeled with [^32^P] using a Rediprime II DNA labeling system (Amersham, Chicago, IL, USA) [35,36]. The Northern blot analyses were repeated twice, showing similar results.

#### 2.5.2. Sp1Transcription Factor (TF)

Northern blot analysis was performed on D.Mel2 cell (2 × 10^5^ cells/mL) that were transfected for 3 h at 27 °C in the presence of Cellfectin with pJHEH#3L3 short promoter (627 bp) cloned in pCaSpeR-AUG-βgal and Sp1 TF dsRNA (363 bp, accession number NM_132351). The Sp1 TF dsRNA was amplified by forward primer DB942 (5′ ATGTTATTGATATTGAAATATAAT 3′) and reverse primer DB943 (5′ TCGTGGTACCAATTGTAGTGTCGATC 3′). After transfection, washing and removal of Cellfectin, the cells were grown for 72 h at 27 °C and the RNA extracted with TRIzol and assayed by Northern blot analysis using a [^32^P] labeled DNA probe of TF Sp1 (363 bp) and [^32^P] labeled actin probe, as discussed above (Section 2.5.1). Control cells were transfected with pJHEH#3L3 promoter, but were not treated with TF Sp1 dsRNA. The Northern blot analysis was repeated twice, showing similar results.

### 2.6. Molecular Modeling of JHEH 1, 2 and 3

Molecular modeling by homology of JHEH 1 (AF517545), JHEH 2 (AF517546) and JHEH 3 (AF517547) from *D. melanogaster* was performed with the YASARA structure program [37], using JHEH from the silkworm *Bombyx mori* (PDB 4QLA) [38], JHEH from *Streptomyces carzinostaticus* subsp. *Neocarzinostaticus* (PDB 4I19 and 5F4Z), epoxide hydrolase from *Aspergillus niger* (PDB 1Q07 and 3G02)) [38,39] and epoxide hydrolase from *Aspergillus* usamii (PDB 6IX4), as templates for the *D. melanogaster* JHEHs. Up to 22, 23 and 24 different models were built for JHEH 1, JHEH 2 and JHEH 3, respectively, leading to a single hybrid model from the different previous models for each JHEH. PROCHECK [40], ANOLEA [41] and the calculated QMEAN scores [42,43], were used to assess the geometric and thermodynamic qualities of the three-dimensional models.

The geometric and thermodynamic qualities of the three-dimensional models are summarized in Table 1. Although modeling of extended loops gives conformations of poorly reliable geometric and thermodynamic qualities, the calculated QMEAN scores nevertheless gave acceptable values of 0.72, 0.75 and 0.70 for the JHEH 1, JHEH 2 and JHEH 3 homodimers models, respectively (Table 1). Docking of JH III and JH IIIA to the 3D model of JHEH 3 was performed with the YASARA structure program. Docking experiments were also performed at the SwissDock web server (http://www.swissdock.ch (accessed on 14 June 2022)) [44,45], to confirm our docking results, and molecular cartoons were drawn with Chimera [45] (Figure 4).

### 2.7. Cloning and Characterization of JHEHs Promoters

The promoter regions of JHEH 1, 2 and 3 were identified by blasting the *D. melanogaster* genome using Lasergene Genomic Suite software (DNASTAR) (see Section 2.4 and Figure 2). Promoter sequences of JHEH 1, 2 and 3 were tested for transcriptional activities by sequentially cutting the promoters into smaller-length sequences using PCR and primers at different 5′ end positions of the promoter and the same primer at the 3′ end. Each primer carried a restriction site enzyme to allow unidirectional cloning into plasmid pCaSpeR-AUG-βgal at the multiple cloning site (Appendix A). The recombinant plasmid (2 μg) carrying the short promoters was transfected into D.Mel2 cells using Cellfectin for 3 h at 27 °C. The medium with the Cellfectin was removed and replaced with new serum-free medium (SFM), the cells were incubated at 27 °C for 72 h and promoter transcriptional activity was tested using a β-galactosidase assay, as described earlier (Section 2.1). JH III, JH IIIA and farnesoic acid in hexane (5 μL) were each added to the medium (1.5 mL) after Cellfectin was removed, and the cells were incubated for 72 h at 27 °C. The same amount of hexane (5 μL) without JH III, JH IIIA or farnesoic acid was added to the control cells. Ecdysone was dissolved in sterile medium and (5 μL) was added to the transfected cells after Cellfectin was removed, and the cells were incubated for an additional 72 h at 27 °C. 

#### 2.7.1. JHEH 1 Promoter Assay

A full-length promoter of 845 bp was identified and cloned into pCaSpeR-AUG-βgal behind *lac*Z (Appendix A). Five smaller-length promoters (645, 446, 305, 245 and 145 bp) were obtained via PCR using forward primers DB737, DB792, DB795, DB817, DB828, DB847 and reverse primer DB738. The forward primers carrying *Eco*RI and reverse primer carrying *Bam*HI cleavage sites allowed unidirectional cloning into pCaSpeR-AUG-βgal behind *lac*Z and transfection into D.Mel2 cells. The transfected cells were assayed for β-galactosidase activity to detect promoters’ transcriptional activities (Appendix A, Appendix A and Figure 5a,b).

#### 2.7.2. JHEH 2 Promoter Assay

A 2.6 kb promoter was identified (Figure 2), and part of the full-length promoter (1325 bp) was cloned into pCaSpeR-AUG-βgal behind *lac*Z (Appendix A, Appendix A and Figure 6a). Five smaller promoter sequences (850, 585, 455, 245 and 146 bp) were obtained via PCR using forward primers DB793, DB796, DB808, DB848, DB860 carrying *Eco*RI cleavage site and reverse primer DB740 carrying *Bam*HI cleavage site, allowing unidirectional cloning into pCaSpeR-AUG-βgal behind *lac*Z. After transfection into *D. melanogaster* cells, the cells were tested for β-galactosidase activity assay to determine promoter transcriptional activity (Appendix A, Appendix A and Figure 6a,b).

#### 2.7.3. JHEH 3 Promoter Assay

A 3.0 kb promoter was identified (Figure 2), and part of the full-length promoter (1562 bp) was cloned into pCaSpeR-AUG-βgal behind *lac*Z (Appendix A, Appendix A and Figure 7a). Six smaller promoter sequences (852, 627, 452, 332, 212, 112 bp) were obtained via PCR using forward primers DB794, DB797, DB809, DB829, DB849 and DB861, each with a *Bam*HI cleavage site and reverse primer DB742 with a *Kpn*I cleavage site, allowing unidirectional cloning into pCaSpeR-AUG-βgal behind *lac*Z. After transfection into *D. melanogaster* cells, the transfected cells were assayed for β-galactosidase activity to determine promoter transcriptional activity (Appendix A, Appendix A and Figure 7a). A promoter sequence (225 bp) between the first and second PCR shorter promoters pJHEH#3L2 and pJHEH#3L3 (Figure 7a) was assayed with different primers (Appendix A and Figure 8) to find out if the DNA sequence contains TF(s)-binding sites that downregulate or upregulate the JHEH 3 promoter’s activity.

### 2.8. Statistical Analysis

Data were analyzed using Student’s *t*-test using GraphPad Prism v5.0. Results were considered statistically significant when *p* < 0.05 and expressed as means of 3 determinations ± SEM., except where otherwise stated.

## 3. Results

### 3.1. cDNA Sequencing of JHEH 1, 2 and 3

Full-length cDNA sequences were generated using PCR and RT-PCR strategies described in the Materials and Methods section (Section 2.4) (Figure 1a–c). The sequences were deposited in the GenBank (accession numbers AF517545, AF AF517546 and AF517547). The cDNA sequences of JHEH 1, 2 and 3 are 1470, 1543 and 1441 bp long, with 1425, 1398 and 1407 bp of open reading frames (ORFs) of 409, 463 and 468 amino acids (Figure 1a–c) and poly adenylation sequences of AATAAA at 1429, 1478 and ATAAA at 1419 for JHEH 1, 2 and 3, respectively. The three JHEH sequences exhibit a typical catalytic groove of D241, D417 and H444 for JHEH 1, D236, E412 and H439 for JHEH 2 and D236 and D412 and H439 for JHEH 3 (Figure 1a–c), including W166, W163 and W163 for JHEH 1, 2 and 3, respectively, which plays an important role in binding the substrate in the active groove of JHEH. To find the promoter regions, Network Promoter Prediction software was used [46] (http://www.fruitfly.org/seq_tools/promoter.html (accessed on 15 June 2022)), predicting promoter regions of 0.84 kb, 2.6 kb and 3.0 kb for *jheh* 1, 2 and 3, respectively. Three introns were found in the *jheh* 1 and 2 exons and 2 intron in the *jheh* 3 exon. The *jheh* sequence is directionally following the *D. melanogaster* genome (Figure 2)

### 3.2. Northern Blot Analyses

To detect *jheh* 1, 2 and 3 transcripts in female *D. melanogaster,* head thoraxes, gut, ovaries, fat bodies and whole females were removed and assayed using Northern blot analyses with *jheh* 1, 2 and 3-specific [^32^P] labeled probes (Appendix A). Northern blot analyses show that *jheh* 1, 2 and 3 transcripts of 1.47, 1.54 and 1.44 kb, respectively, are found in the head thorax, gut, ovary, fat bodies and whole insect RNA extracts. These results show that *jheh* 1, 2 and 3 transcripts are synthesized in many tissues (Figure 3a–c). Because JH has many physiological targets, and JHEH is the main enzyme that irreversibly inactivates JH III, it is found to be ubiquitously distributed. The densities of the *jheh* 1 and 3 transcript bands after Northern blot analyses are similar, whereas the *jheh* 2 transcript bands are 3–4-fold denser, indicating that more *jheh* 2 transcript is synthesized by female *D. melanogaster* head thorax, gut, ovary and fat body compared with *jheh* 1 and 3 transcripts (Figure 3a–c). Northern blot analyses of the *act* transcript indicate that the transfer was even in all the lanes.

### 3.3. Molecular Modeling of JHEH 1, 2 and 3

The three-dimensional models built for the three JHEH homodimers JHEH 1, JHEH 2 and JHEH 3 of *D. melanogaster* are very similar (Figure 4a–c). They consist of two dimers associated by non-covalent bonds as a homodimer. Each monomer conformation predominantly contains α-helices associated with a single β-sheet structure. Because of these structural similarities, JHEH 1, JHEH 2 and JHEH 3 are closely superimposed on each other, with a root mean square deviation (RMSD) of 0.87 A between 381 pruned atom pairs corresponding to the structurally conserved α-helices and β-sheets of the three models (Figure 4d). A catalytic groove containing the catalytic triad D236, D412 and H439 occurs at the surface of each monomer. The H161 G162 W163 P164 motif characteristic of the epoxide hydrolases enzymes, and a tyrosine residue (Y382) occurs on the edge of the catalytic groove (Figure 4e,f). In each monomer, the catalytic groove is well exposed and leads to a depression that separates both monomers in the homodimeric structure. Docking experiments performed with JH III and JH IIIA resulted in the anchoring of both hormones to the catalytic groove of JHEH 3 via a network of four and seven hydrogen bonds, respectively (Figure 4e,f). Both hormones are similarly linked to the amino acid residues D236 and D412 of the catalytic triad, and to Y382 located on the edge of the catalytic groove. The linkage of JH III and JH IIIA only differs by the number of hydrogen bonds connecting the epoxide group to D236 (a single H-bond for JH III compared with two H-bonds for JH IIIA) and the acidic group to D412 (two H-bonds for JH III compared with four H-bonds for JH IIIA), showing a different accommodation of both hormones in the catalytic groove of JHEH 3 (Figure 4e,f). Therefore, JH IIIA shows a better affinity for JHEH 3 compared with JH III.

### 3.4. Activities of JHEH 1, 2, 3 Promoters

To follow the activity of each JHEH promoter, shorter promoter sequences were amplified by PCR (Appendix A) and cloned into plasmid pCaSpeR-AUG-βgal at the multiple cloning site behind *lac*Z (Appendix A), and the β-galactosidase transcriptional activity driven by each truncated promoter sequence was assayed (Section 2.8)

#### 3.4.1. JHEH 1 Promoter

Shorter JHEH 1 promoter sequences were compared with the uncut promoter, showing that the full promoter sequence (845 bp, pJHEH#1) and a shorter promoter sequence (645 bp, pJHEH#1L1) exhibit similar activities (Figure 5a,b). On the other hand, much shorter promoter sequences pJHEH#1L2, pJHEH#1L3, pJHEH#1L4 and pJHEH#1L5 (445, 305, 245 and 145 bp, respectively) exhibited significantly lower activities (*p* < 0.05) of 1.4, 1.8, 1.7 and 7.5-fold, respectively, compared with the full-length promoter. The lowest activity was observed when the promoter’s sequence was 145 bp long (Figure 5a,b). D.Mel2 cells that were not transfected or transfected with an empty plasmid (pCaSpeR-AUG-βgal) did not exhibit β-galactosidase activity, indicating that the enzymatic activities observed were due to JHEH 1 promoter sequences driving *lac*Z.

#### 3.4.2. JHEH 2 Promoter

JHEH 2 promoter sequence (1325 bp, pJHEH#2L1) (Figure 6a) was cloned into plasmid pCaSpeR-AUG-βgal and expressed in D.Mel2 cells and β-galactosidase activity of 5.8 mU was determined in cells that were transfected with pJHEH#2L1 promoter (Figure 6b). Shorter promoters pJHEH#2L2 and pJHEH#2L3 (850 and 585 bp, respectively, Figure 6a) expressed β-galactosidase activities that were not significantly different than pJHEH#2L1 (1325 bp) (results not shown). When the promoter length was shortened from 455 bp to 245 bp (pJHEH#2L4 and pJHEH#2L5, respectively) (Figure 6a) the β-galactosidase activity significantly increased (*p* < 0.05) from 6.2 mU to 10.6 mU. No β-galactosidase activity was observed when a shorter promoter sequence (146 bp, pJHEH#2L6) was tested (Figure 6a,b). Control D.Mel2 cells that were not transfected or transfected with an empty pCaSpeR-AUG-βgal did not exhibit β-galactosidase activity, indicating that the observed enzymatic activities were due to JHEH 2 promoter sequences driving *lac*Z.

#### 3.4.3. JHEH 3 Promoter

JHEH 3 promoter sequences 1562 bp and 852 bp (pJHEH#3L1 and pJHEH#3L2, respectively (Figure 7a), were each cloned into plasmid pCaSpeR-AUG-βgal and expressed in D.Mel2 cells, and β-galactosidase activities of 19 mU and 22 mU were determined in cells that were transfected with pJHEH#2L1 and pJHEH#3L2 promoters (Figure 7b). When the promoter sequence was shortened to 627 bp (pJHEH#3L3, Figure 7b), the β-galactosidase activity expressed in D.Mel2 cells increased by 4.8-fold to 106 mU. Shortening the promoter sequence to 452, 332, 212 and 112 bp and expressing the short promoter sequences in D.Mel2 cells reduced the β-galactosidase activity to 23 mU, 18 mU, 5 mU and 1 mU, respectively (pJHEH#3L4, pJHEH#3L5, pJHEH#3L6, pJHEH#3L7, Figure 7b). The increase in β-galactosidase activity expressed by promoter pJHEH#3L3 is significantly higher (*p* < 0.05) than the longer and shorter segments of the JHEH 3 promoter sequences that were tested (Figure 7a,b). These results prompted us to examine the JHEH 3 promoter sequence between pJHEH#3L2 and pJHEH#3L3 (225 bp, Figure 8a) for TFs or inhibitory sequences. Several TFs were identified (bHLH-CS, H2B-CCAAT, NF-E1-CS2, CAAP-site, TCF-1 and sp1) (Figure 8a). JHEH 3 promoter sequences between pJHEH#3L2 and pJHEH#3L3 (Figure 8a) were amplified as discussed above, using forward and reverse primers (Appendix A) to exclude some of the TF or mutate them by changing the promoter’s nucleotides sequence or removing a whole segment of nucleotides (DB876 and DB877, respectively (Figure 8a). The amplified promoter sequences carrying restriction enzyme sequences at the 5′ and the 3′ ends *Bam*HI and *Kpn*I, respectively (Appendix A), were cloned into pCaSpeR-AUG-βgal. The β-galactosidase activity showed a two-fold increase in activity when DB 843, DB875, DB876 and DB877 (Figure 8a–c) were used compared with full length pJHEH#3L2. However, the increase in activity was still 2 to 2.5-fold lower than the activity of pJHEH#3L3. These results indicate that TF Sp1 plays a major role in activating the pJHEH#3L3 promoter sequence (627 bp) ( Figure 7 and Figure 8). In control D.Mel2 cells that were not transfected or transfected with an empty pCaSpeR-AUG-βgal plasmid, no β-galactosidase activity was found, indicating that the observed enzymatic activities were due to JHEH 3 promoter sequences driving *lac*Z.

**Figure 8 biomolecules-12-00991-f008:**
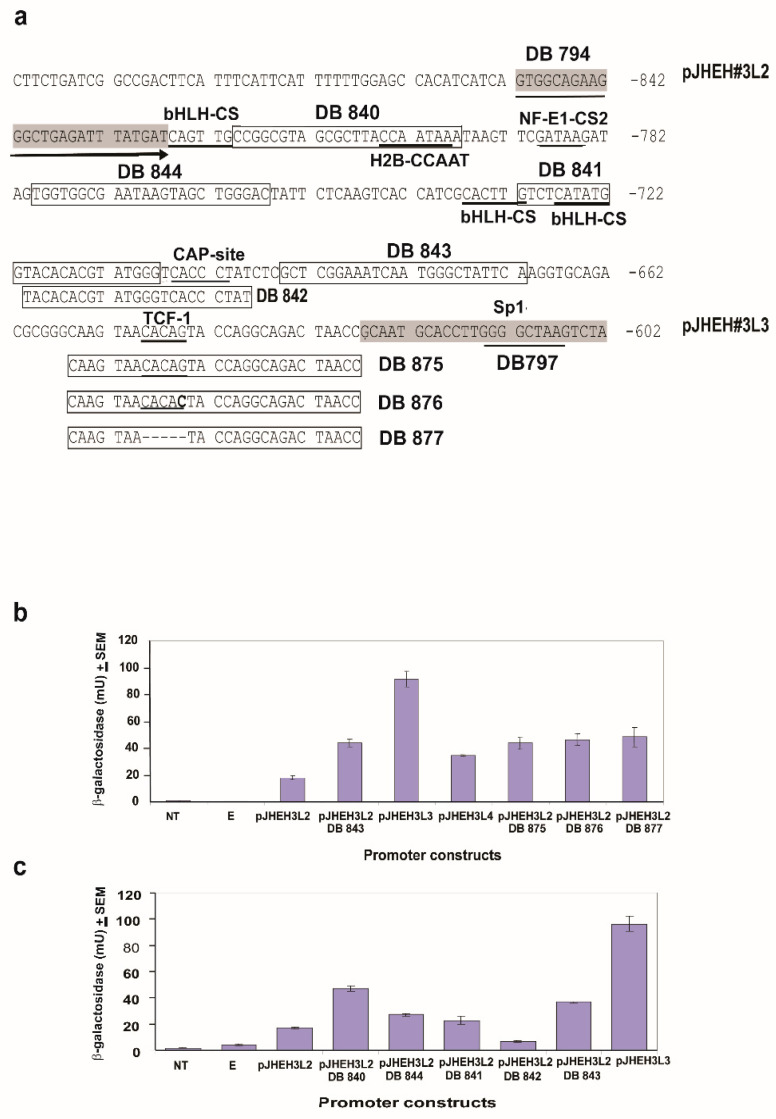
Testing promoter sequence (225 bp) between pJHEH#3L2 and pJHEH#3L3 (**a**) for TFs DNA-binding sequences and effects on *jheh 3* promoter transcriptional activities, as shown in Figure 7, by testing different promoter lengths and mutating or deleting TF TCF-1 DNA-binding site using primers DB877 and DB876, respectively (**b**,**c**).

#### 3.4.4. Northern Blot Analysis and dsRNA Knockdown of TF Sp1

To find out if TF Sp1 plays a major role in the activation of promoter pJHEH#3L3, Sp1 dsRNA was co-transfected with pCaSpeR-AUG-βgal carrying promoters pJHEH#3L2 or pJHEH#3L3 (materials and methods Section 2.5.2). The D.Mel2 transfected cells were grown for 72 h and analyzed using Northern blot analysis for Sp1transcript and β-galactosidase activity. Northern blot analysis of D.Mel2 cells that were transfected with pJHEH#3L3 and not treated with dsRNA (control) detected a TF Sp1 transcript above 3.0 kb, whereas cells that were transfected with dsRNA showed degraded, shorter transcripts at 2.5 kb and 1.5 kb (Figure 9a, right lane). The similar actin bands in each lane indicate an even transfer of the transcripts by the Northern blot (Figure 9a). For cells that were transfected with pCaSpeR-AUG-βgal carrying promoters pJHEH#3L2 or pJHEH#3L3 and not incubated with dsRNA, their β-galactosidase transcriptional activity was 41 and 78 mU, respectively. On the other hand, for cells that were transfected with pJHEH#3L2 and pJHEH#3L3 in the presence of dsRNA, their β-galactosidase transcriptional activity was low (1 and 2 mU, respectively) (Figure 9b). *Drosophila* cells that were not transfected or transfected with empty plasmid (Controls) expressed very low β-galactosidase activity (Figure 9b). These results indicate that TF Sp1 plays an important role in upregulating JHEH 3 promoter.

### 3.5. The Effects of JH III, JH IIIA, 20HE and Farnesoic Acid on JHEH 3 Promoter

To find out the effects of JH III and JH IIIA on JHEH 3 promoter sequences (pJHEH#3L1, pJHEH#3L2, pJHEH#3L3, pJHEH#3L4, pJHEH#3L6, Figure 7a), JH III (1 μM) and JH IIIA (5 μM) were incubated with D.Mel2 transfected cells. JH III significantly (*p* < 0.05) inhibited all the promoter constructs compared with hexane-treated controls (Figure 10a), whereas JH IIIA did not significantly affect the activity of the JHEH 3 promoter sequences that were tested, including JHEH 1 full promoter (pJHEH#1, 845 bp, Figure 10a), compared with controls treated with hexane (Figure 10b). Incubations with 20HE (1 and 5 μM) showed that 20HE is a potent inhibitor of JHEH 3 promoter pJHEH#3L3 (Figure 11a,b), significantly (*p* < 0.05) inhibiting the transcriptional activity of the promoter by 2 and 3.8-fold in the presence of 1 and 5 μM 20HE, respectively, and significantly (*p* < 0.05) inhibiting the promoter by 2.7-fold in the presence of 20HE mimic (RH59992) (Figure 11b). Cells that were not transfected or transfected with empty plasmid were not affected (Figure 11a,b). On the other hand, farnesoic acid (1μM) did not significantly affect the JHEH 3 promoter sequences (pJHEH#3L1, pJHEH#3L2, pJHEH#3L3, pJHEH#3L4 and pJHEH#3L6) transcriptional activities (Appendix A).

### 3.6. Effect of JHEH 3 pJHEH#3L3 Promoter on Transgenic D. Melanogaster

Our Northern blot analyses showed that JHEH transcripts are found in the head thorax, gut, ovary, the fat body and the whole-body extract of female *D. melanogaster.* Therefore, the most active JHEH promoter pJHEH#3L2 (627 bp, Figure 7a) was cloned into pCaSpeR-AUG-βgal and injected with a helper plasmid into *D. melanogaster* embryos. Transformed flies that were assayed for β-galactosidase activity and observed under a dissecting microscope exhibited enzymatic activity in many parts of the transgenic flies, in the abdomen, thorax, leg muscle and the junction between the abdomen and thorax (Figure 12), indicating that the JHEH 3 promoter is active in many *D. melanogaster* tissues, confirming our Northern blot analysis (Figure 3c). Control flies that were transformed with an empty plasmid did not show transcriptional activity of β-galactosidase (results not shown).

## 4. Discussion

We identified and sequenced the cDNAs of JHEH 1, 2 and 3 from female *D. melanogaster* using extracted RNA and a λ Zap cDNA library. The cDNAs code for proteins of 474, 463 and 468 amino acids, respectively (Figure 1a–c). The proteins contain XWG anchor motif (W40, W41 and G42 for JHEH 1; Y41, W42 and G43 for JHEH 2 and Y42, W43 and G44 for JHEH 3) (Figure 1a–c), which is involved in subcellular localization of JHEH in *Bombyx mori*, *Lymantria dispar* and *Apolygus lucorum*, and is highly conserved in higher organisms, but absent in fungi, bacteria and protozoa [38,47,48]. JHEH 1, 2 and 3 amino acid sequences contain a HGWP motif and W166, W163 and W163 as part of the motif in JHEH 1, 2 and 3, respectively (Figure 1a–c). The catalytic amino acids triad of JHEH 1, and 3 is D, D, H and for JHEH 2 D, E, H (Figure 1a–c). Substitution of D with E is found in the late trypsin of *Aedes aegypti*, E196 replaced D at the specificity pocket of *A. aegypti* late trypsin without affecting the enzyme’s activity [49]. The *jheh* of *D. melanogaster* sequence exhibits three related genes, *jheh 1, jheh 2 and jheh 3,* with introns and promoters of different lengths (0.84, 2.6 and 3.0 kb, respectively) (Figure 2). The *jheh 2* of *D. melanogaster* larvae was suggested to function as a microsomal Epoxide Hydrolase (mEH) that does not participate in the JH III metabolism but works in concert with other xenobiotic metabolizing enzymes [20]. It is interesting to note that these authors also reported that a whole-body extract of *D. melanogaster* larva converted JH III into its diol, indicating that *D. melanogaster* larvae have functional JHEHs that can metabolize JH III. The exact function(s) of JHEH 2 in adult *D. melanogaster* are yet to be determined. Northern blot analyses show that JHEH 1, 2 and 3 transcripts are found in the head thorax, gut, ovary, fat body and whole extract of females, confirmed by our sequencing data, which show that JHEH 1, 2 and 3 contain membrane anchoring sequences that allow the enzymes to be anchored to membranes of many tissues of female *D. melanogaster*. It is interesting to note that the intensities of JHEH 2 transcript bands are much higher than the other transcript bands of JHEH 1 and JHEH 3 (Figure 3a–c), indicating that perhaps more JHEH 2 is synthesized by female *D. melanogaster*, and that the enzyme may have a dual function as a JH III metabolizing enzyme and also as a mEH [20]. Three-dimensional molecular models of the three JHEHs show that the proteins are associated as homodimers by non-covalent bonding, and are made up primarily of α-helices associated with a β-sheet showing high similarity, which allows them to be superimposed on each other (Figure 4a–d). A homodimer structure for JHEH has been determined by X-ray crystallography for the silkworm *Bombyx mori* [16]. Our 3D models predict that JHEH 1, 2 and 3 can be superimposed, as they have similar spatial conformations. To study the substrate specificity of JHEH 3, we docked JH III and JH IIIA into the catalytic groove of JHEH 3. The binding of the two substrates to JHEH 3 is different; JH IIIA binds stronger than JH III to JHEH 3, using two H-bonds at the epoxide group to bind to D236 compared with one for JH III, and four hydrogen bonds to bind to D412 compared with two H-bonds for JH III (Figure 4g,h). Zhou et al. [16] used X-ray crystallography to show that JH II binds in the active groove of *B. mori* JHEH using two H-bonds at the epoxide group and one H-bond at D387 in their active site; however, they did not use JH IIIA in their studies. Our results indicate that the binding of JHEH III to JH IIIA is much tighter than the binding to JH III, and therefore, JH IIIA is probably the preferred JHEH substrate and not JH III. To find out if JHEH(s) in female *D. melanogaster* preferred substrate is JH IIIIA, we treated female *D. melanogaster* at different times after adult eclosion (3, 24, 48 and 72 h) with [12-^3^H]JH IIIA and separated the radioactive-labeled metabolites 1 h after the treatment using reversed phase C_18_ HPLC [6]. The ratio between JH IIIA and JH IIIAD at 3, 24, 48 and 72 h after adult eclosion was 2.3, 1.25, 0.79 and 0.85, respectively (Table 2), indicating that female *D. melanogaster* JHEH(s) metabolize JH IIIA into JH IIIAD, confirming our 3D molecular modeling (Figure 4h).

In mosquitoes, it was shown that JH IIIA is also the preferred substrate for JHEH. Treating female *A. aegypti* with [12-^3^H](10R)-JH III and analyzing the extracts using C_18_ reversed phase HPLC determined the ratio between JH IIIAD, JH IIIA and JH IIID as 17/4/1, and treating female *A. aegypti* with [12-^3^H]JH IIIA converted 50% of the initial JH IIIA in 1 h into JH IIIAD [6]. The JH III titer in *D. melanogaster* rapidly declines after adult eclosion [50] and JHEH, in concert with JHE play an important role in JH III metabolism. To understand how JHEH 1, 2 and 3 are controlled by their promoters, we reduced the length of the promoter sequences and cloned the short sequences into plasmid pCaSpeR-AUG-βgal (Appendix A) and transfected D.Mel2 cells. Full-length JHEH 1 promoter (845 bp, pJHHE#1) exhibited the highest transcriptional activity, whereas shorter promoter sequences reduced the transcriptional activity, and the shortest promoter (145 bp, pJHEH#1L5) exhibited the lowest transcriptional activity, which was 5.6-fold lower than pJHEH#1 (Figure 5a,b). Similar approaches were used to study the anhydrobiotic midge, *Polypedilum vanderplanki* 121 promoter that allows survival under desiccated conditions [24], as well as to identify and functionally analyze heat shock promoters from *S. frugiperda*. [25]. JHEH 2 promoter pJHEH#2L1 (1325 bp), on the other hand, showed that shorter promoter length increased the promoter’s transcriptional activity, reaching a maximum at a promoter length of 245 bp (pJHEH#2L5, Figure 6a,b). These results indicate that shortening the promoter removed DNA sequences that were used as TF(s)-binding sites to downregulate the transcriptional activity of the promoter [23,24]; however, a shorter promoter of 148 bp was not active. The effect of TF(s)-binding sites on the JHEH 3 promoter’s transcriptional activity was studied by shortening the promoter and assaying its transcriptional activity. The initial promoter length that we studied (pJHEH#3L1 1562 bp, Figure 7) exhibited transcriptional activity (24 and 10-fold) higher when compared with JHEH 1 and 2 promoters (Figure 5 and Figure 6). A shorter promoter sequence (pJHEH#3L3, 627 bp, Figure 7) exhibited higher transcriptional activity than all the other promoter sequences (Figure 7a,b), indicating that the sequence between pJHEH#3L2 and pJHEH#3L3 (225 bp) may have sequences that bind TF(s) that downregulate or upregulate the promoter. The promoter sequence between pJHEH#3L2 and pJHEH#3L3 contains several TF(s)-binding sites (bHLH-CS, HZB-CCAAT,CAP-site, TCF-1 and Sp1). All the TFs DNA-binding sites were between pJHEH#3L2 and pJHEH#3L3 except for the Sp1 DNA-binding site, which is located at the 5′ end of pJHEH#3L3 (Figure 7a). Analysis of the promoter TF(s)-binding sites indicates that only a two-fold increase in promoter transcriptional activity was observed when several of the promoter DNA sequences were shortened and mutated, especially the DNA sequence that binds TF TCF-1 (Figure 8a,b). The highest promoter’s transcriptional activity was observed with promoter pJHEH#3L3, indicating that TF Sp1 upregulates that promoter. TF Sp1 is involved in basal transcriptional regulation of various genes that are involved in many cellular processes, including cell differentiation, cell growth, apoptosis, immune responses, response to DNA damage and chromatin remodeling. Post-translational modifications such as phosphorylation, acetylation, glycosylation and proteolytic processing significantly affect the activity of this protein, which can act as an activator or a repressor [51]. Knocking down Sp1 in D.Mel2 cells that were transfected with pJHEH#3L2 and pJHEH#3L3 by dsRNA stopped both promoters’ transcriptional activities and degraded the Sp1 transcript (Figure 9a,b). These results show, for the first time, that the role of Sp1 is to activate the JHEH 3 promoter. JH metabolism is an important aspect of insect development [1,2] during in the life of male and female *D. melanogaster* [52]. The activities of JHEH 3 promoter sequences that were tested in the presence of JH III (1 μM) were significantly inhibited compared with controls (*p* < 0.05), whereas incubation with JH IIIA at five-fold higher concentration (5 μM) did not significantly inhibit transcriptional activities of the tested promoter sequences (Figure 10a,b). These results indicate that JH IIIA is the preferred substrate for JHEH 3 and not JH III. In insects, larval–larval molting and larval–pupal–adult metamorphosis are elicited by 20 HE [50,53]. During this period, the level of JH III in *Drosophila* is low, whereas the level of 20HE is high [52]. A relationship between JH III and 20HE exists, in which JH III stays at a low level, and it is not advantageous for the insect to metabolize JH III at this time. We tested this hypothesis by incubating 20HE and 20HE mimic RH5992 with the most active promoter’s sequence (pJHEH#3L3, Figure 7a,b) of JHEH 3. We showed that 20HE at low concentrations (1 and 5 μM) and its mimic (5 μM) significantly inhibited the JHEH 3 promoter’s activity, suggesting that in *D. melanogaster* 20HE inhibits the activity of JHEH 3 promoter, and probably the activities of JHEH 1 and 2 promoters. However, the suggested effect of 20HE on JHEH 1 and 2 promoters needs to be tested in future work. Farnesoic acid (1 μM), which is several steps away in the biosynthetic pathway of JH III, did not have an effect on the transcriptional activity of any of the JHEH 3 promoter sequences (Appendix A). Our Northern blot analyses of JHEH 1, 2 and 3 transcripts show that the transcripts are ubiquitously expressed in many tissues (head thorax, gut, ovary fat body and the whole insect). To find out where JHEH 3 promoter is active, transgenic *D. melanogaster* that were transformed with JHEH 3 promoter (pJHEH#3L3, 627 bp, Figure 7a) were cloned in pCaSpeR-AUG-βgal, tested for β-galactosidase activity and observed under a dissecting microscope. β-galactosidase activity was found in the abdomen, thorax, leg muscle and the junction between the abdomen and thorax (Figure 12a–d), confirming that the JHEH 3 promoter and its transcript are located in many tissues. These results indicate that JHEH 3, and probably JHEH 1 and 2 promoters, are distributed throughout *D. melanogaster,* and their function is to control the metabolism of JH III in many of its target tissues. Our results also indicate that 20HE and JH III play an important role in the control of JHEH 3 promoter. When the 20HE titer is high during adult stages and larval stages, it inhibits JHEH III promoter (Figure 13a). During the initial metabolism of JH III, JHEH does not convert JH III into the JH IIID because JH III inhibits the transcriptional activity of the JHEH promoter (Figure 13a).

After JHE converts JH III into JH IIIA, only then does JHEH promoter, in concert with Sp1, transcriptionally activate *jheh* 3, synthesizing JHEH 3 that converts JH IIIA into JH IIIAD (Figure 13b). This sequence of events is similar to what was shown for female *A. aegypti* [6]. In larval *D. melanogaster* cells, it was shown that *jhe* is stimulated by 1 μM JH III and inhibited by 1 μM 20HE [54], supporting our results that JHEH does not act on JH III because it inhibits the *jheh* promoter. These results, together with our 3D molecular modeling (Figure 4g,h) suggest that JH IIIA is the preferred substrate of JHEH 3 and *jheh* promoter is controlled by JH III and is activated by TF Sp1.

## 5. Conclusions

Our results show that JHEH 1, 2 and 3 promoter sequences exhibit transcriptional activities in transfected D.Mel2 cells. Testing of the JHEH 3 promoter identified, for the first time, a TF Sp1 DNA-binding sequence that upregulates the promoter and plays an important role in activating the JHEH 3 promoter of *D. melanogaster*. A highly transcriptional active promoter sequence of JHEH 3 (627 bp) was shown to be active in many tissues of transgenic *D. melanogaster,* indicating that JHEH 3 is ubiquitously distributed in many tissues. 

## Figures and Tables

**Figure 1 biomolecules-12-00991-f001:**
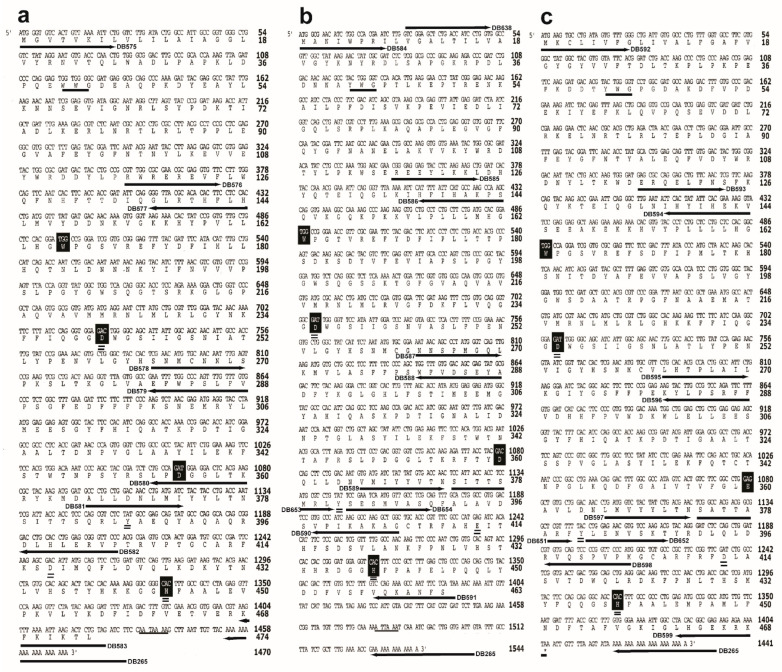
*jheh* 1, 2 and 3 cDNA cloning strategies and primers (arrows) that were used in the cloning and the sequencing. (**a**). *jheh* 1 cDNA amino acids and nucleotides sequence, the membrane anchor sequence WWG is single underlined, the catalytic amino acids triad DYD is double underlined, the H that participates in the catalytic activity is double underlined and the polyadenylation signal sequence AATAAA is single underlined. (**b**). *jheh* 2 cDNA amino acids and nucleotides sequence, the membrane anchor sequence YWG is single underlined, the catalytic amino acids triad DYE is double underlined, the H that participates in the catalytic activity is double underlined and the poly adenylation signal ATTAAA is single underlined. (**c**). *jheh* 3 cDNA amino acids and nucleotides sequence, the membrane anchor sequence YWG is single underlined, the catalytic amino acids triad DYD is double underlined, the H that participates in the catalytic activity is double underlined and the poly adenylation signal is not shown.

**Figure 2 biomolecules-12-00991-f002:**
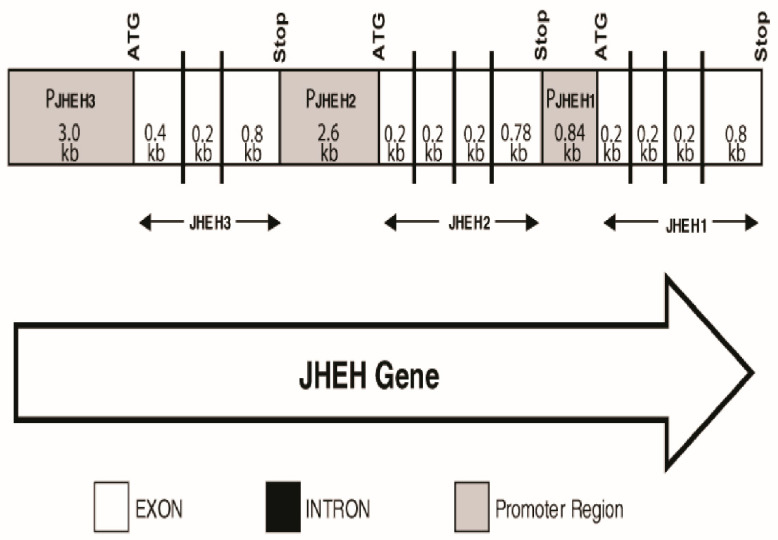
Genomic DNA sequence of *jheh* including intron (black lines) exons (white colored squares) and promoters (gray colored squares) of *jheh* 1, *jheh* 2 and *jheh* 3. Arrow follows the 5′ to 3′direction of *D. melanogaster* genome.

**Figure 3 biomolecules-12-00991-f003:**
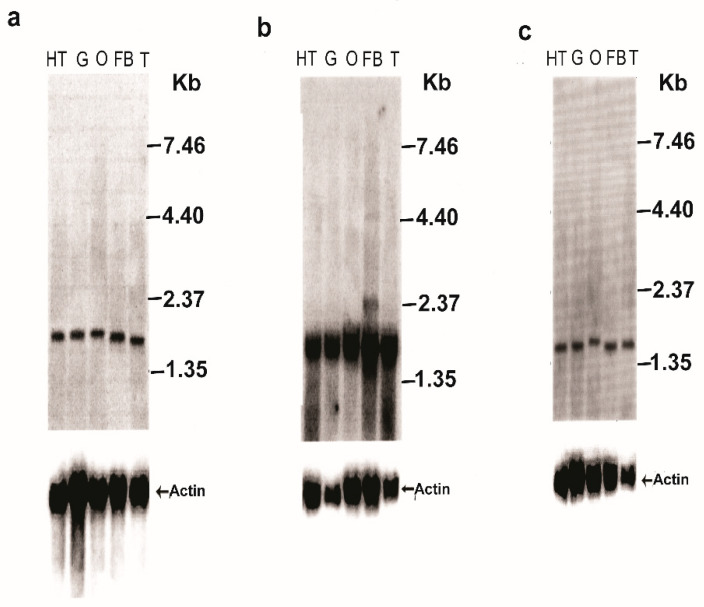
Northern blot analyses of *D. melanogaster jheh* 1 t (**a**), *jheh* 2 (**b**) and *jheh* 3 (**c**) transcripts in the head gut (HT), gut (G), ovary (O), fat body (FB) and total insect extract (T) of female *D. melanogaster*. Blots were hybridized with specific probes for each transcript (Appendix A). Actin probe was used to show even transfer to the blot. The Northern blot analyses were repeated twice, showing similar results.

**Figure 4 biomolecules-12-00991-f004:**
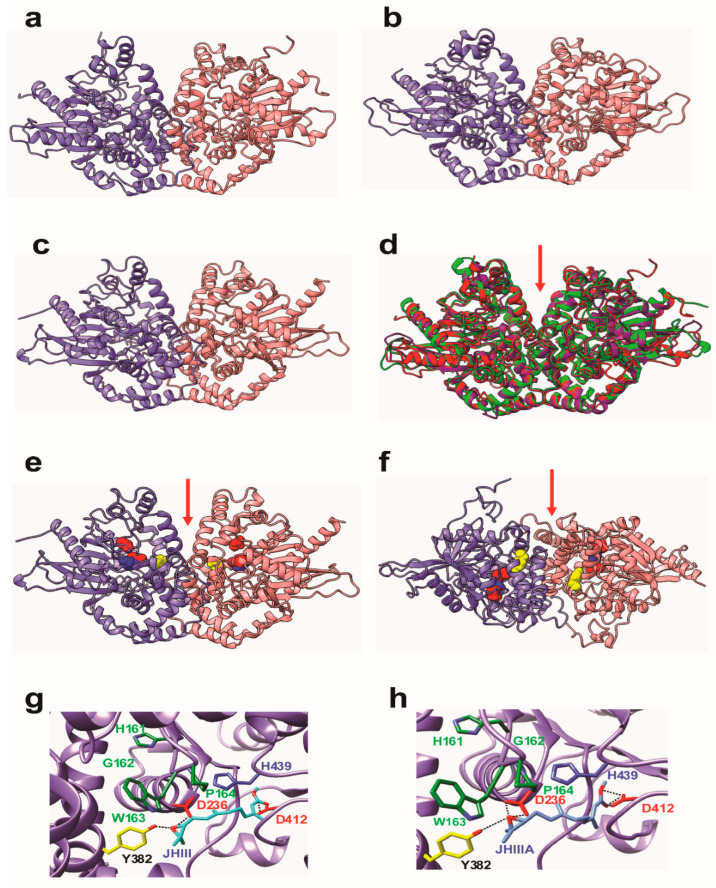
3D models showing non-covalently associated homodimeric monomers of (**a**) JHEH 1 (**b**) JHEH 2 and (**c**) JHEH 3. The two monomers are colored violet and pink, respectively. (**d**) Superpositioning of the 3 models built for JHEH 1 (red), JHEH 2 (blue) and JHEH 3 (green) showing good superposition of the α-helices and β-sheets forming the backbones of the homodimers. The red arrow indicates the depression separating both monomers, associated with the catalytic grooves of both monomers. (**e**) Lateral view and (**f**) upper view of JHEH 3 homodimer showing the localization of residues D236 (red), Y382 (yellow), D412 (red) and H439 (blue) forming the catalytic grooves located on either side of the central depression (red arrow). (**g**). Docking of juvenile hormone III (JH III) (colored cyan) to the catalytic groove of JHEH 3. Amino acid residues D236, D412 and H439 forming the catalytic triad of JHEH 3 are colored red and blue, respectively. Tyrosine residue Y382, forming a H-bond with the epoxide group of JH III, is colored yellow. Amino acid residues from the HGWP motif are colored dark green. Hydrogen bonds are indicated by black dashed lines. (**h**). Docking of juvenile hormone III acid (JH IIIA) (colored blue), to the catalytic groove of JHEH 3. Amino acid residues D236, D412 and H439 forming the catalytic triad of JHEH 3 are colored red and blue, respectively. Tyrosine residue Y382, forming a H-bond with the epoxide group of JH IIIA, is colored yellow. Amino acid residues from the HGWP motif are colored dark green. Hydrogen bonds are indicated by black dashed lines. Note that only 2 H-bonds out of total 4 H-bonds of D412 with JH IIIA can be seen at this viewing angle.

**Figure 5 biomolecules-12-00991-f005:**
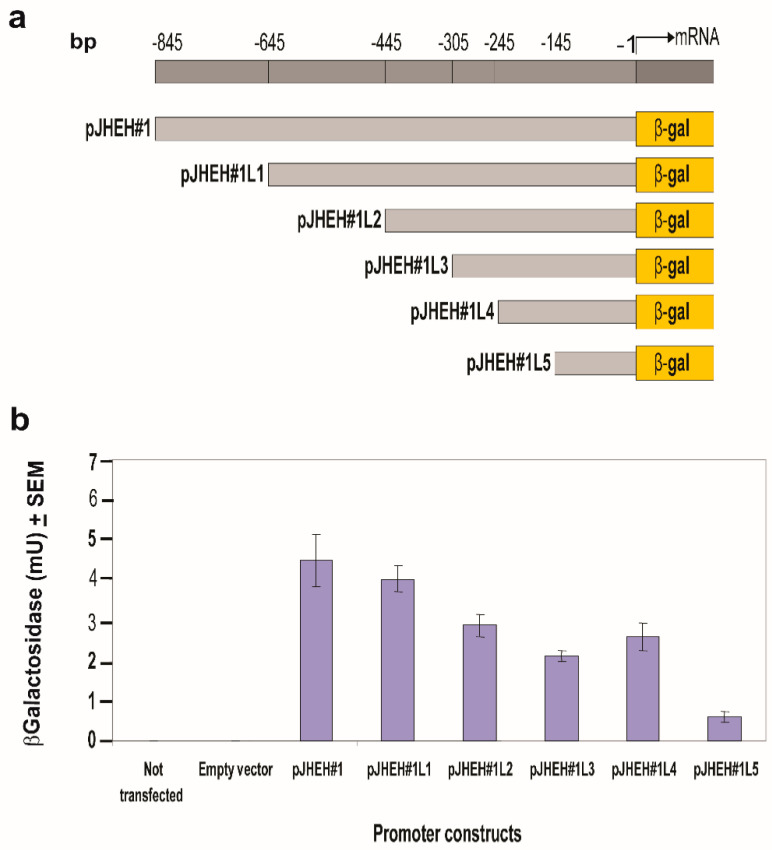
Testing *jheh* 1 different promoter sequence lengths (**a**) for transcriptional activity (**b**) by cloning them into plasmid pCaSpeR-AUG-βgal, transfecting D.Mel2 cells with the recombinant plasmid and assaying 2 × 10^5^ cells for β-galactosidase activity expressed in milliunits (mU). Controls were D.Mel2 cells that were not transfected or transfected with empty plasmid.

**Figure 6 biomolecules-12-00991-f006:**
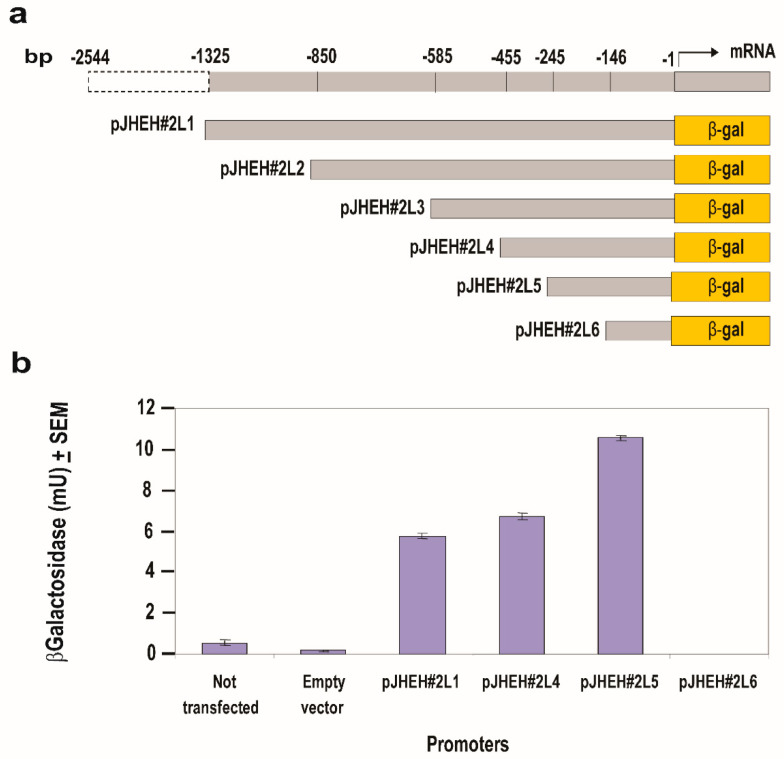
Testing *jheh* 2 different promoter sequence lengths (**a**) for transcriptional activity (**b**) by cloning them into plasmid pCaSpeR-AUG-βgal, transfecting D.Mel2 cells with the recombinant plasmid and assaying 2 × 10^5^ cells for β-galactosidase activity expressed in milliunits (mU). Controls were D.Mel2 cells that were not transfected or transfected with empty plasmid.

**Figure 7 biomolecules-12-00991-f007:**
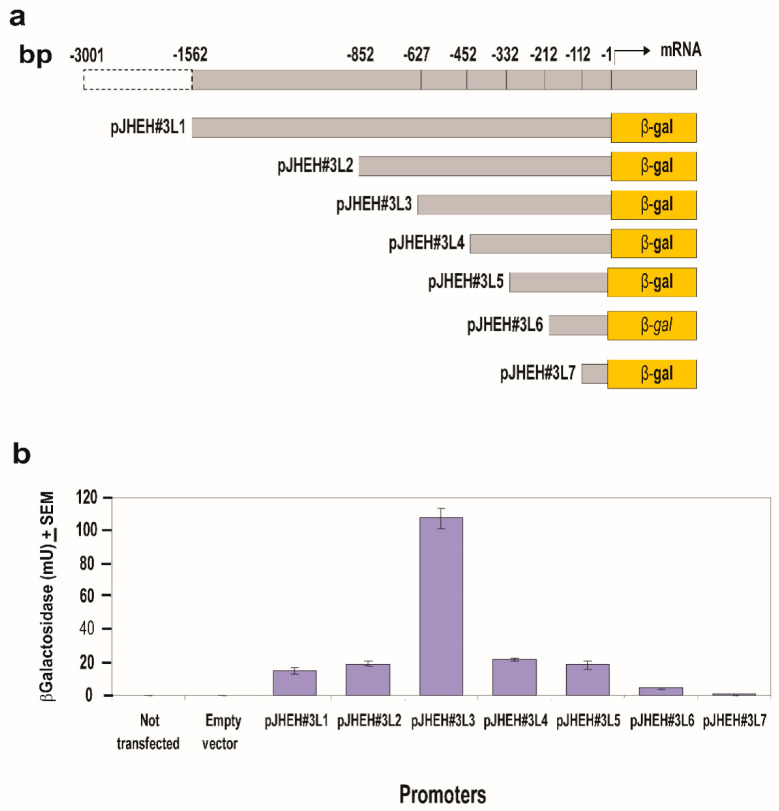
Testing *jheh* 3 different promoter sequence lengths (**a**) for transcriptional activity (**b**) by cloning them into plasmid pCaSpeR-AUG-βgal, transfecting D.Mel2 cells with the recombinant plasmid and assaying 2 × 10^5^ cells for β-galactosidase activity expressed in milliunits (mU). Controls were D.Mel2 cells that were not transfected or transfected with empty plasmid.

**Figure 9 biomolecules-12-00991-f009:**
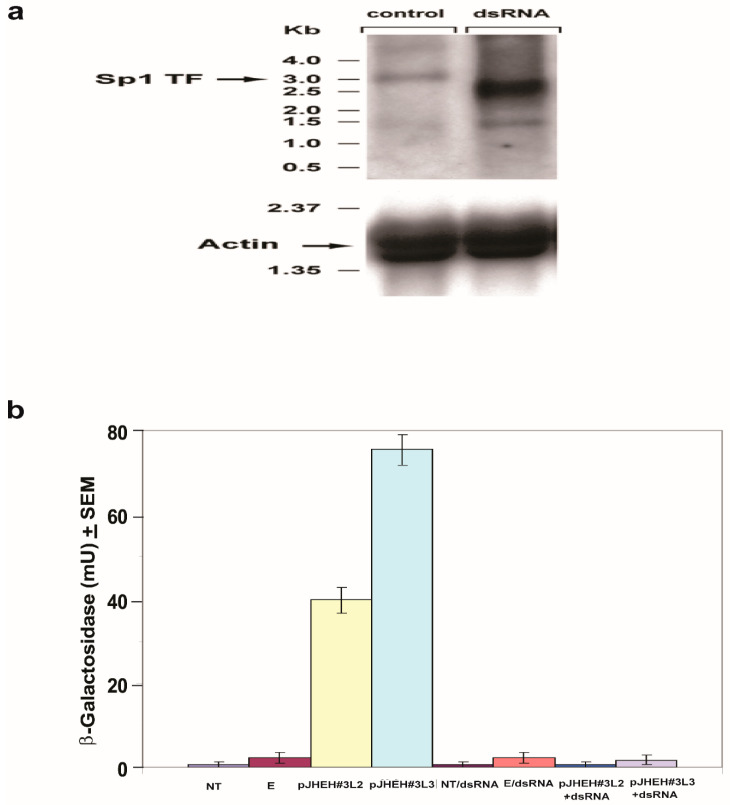
Northern blot analysis of TF Sp1 in D.Mel2 cells that were transfected with promoter pJHEH#3L3 and treated with dsRNA against Sp1. (**a**) In cells that were treated with dsRNA, the Sp1 transcript is degraded (right lane) compared with untreated cells (left lane). (**b**) Transcriptional activity of pJHEH#3L2 and pJHEH#3L3, in transfected D.Mel2 cells is not affected, whereas the activity in cells that were treated with dsRNA is similar to cells that were not transfected (NT) or cells that were transfected with an empty plasmid ©.

**Figure 10 biomolecules-12-00991-f010:**
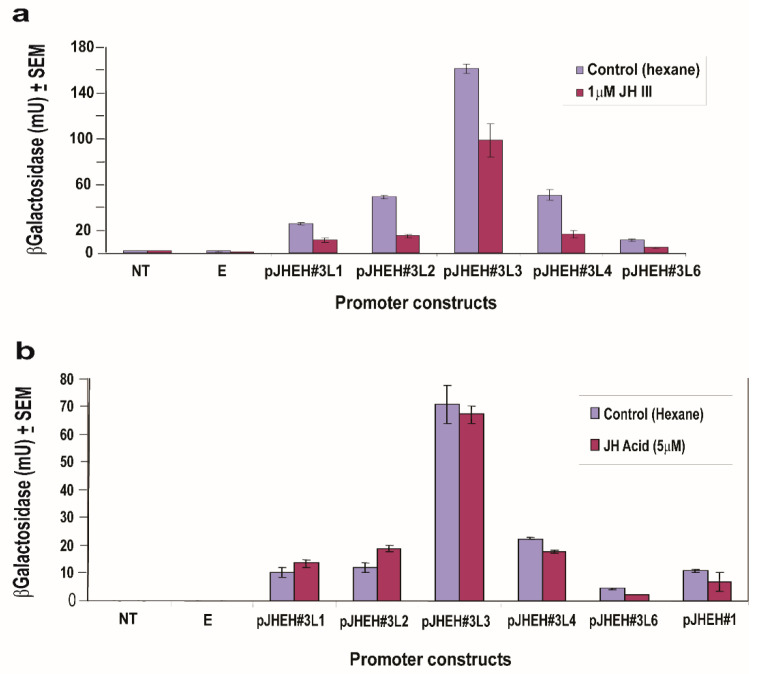
Transcriptional activities of different *jheh* 3 promoter sequences in transfected D.Mel2 cells in the presence of JH III (1 μM) (**a**) and in the presence of JH IIIA (5 μM) (**b**). The different promoter sequence lengths are found Figure 8 and Figure 6. Non-transfected cells (NT) and cells transfected with empty plasm © (E).

**Figure 11 biomolecules-12-00991-f011:**
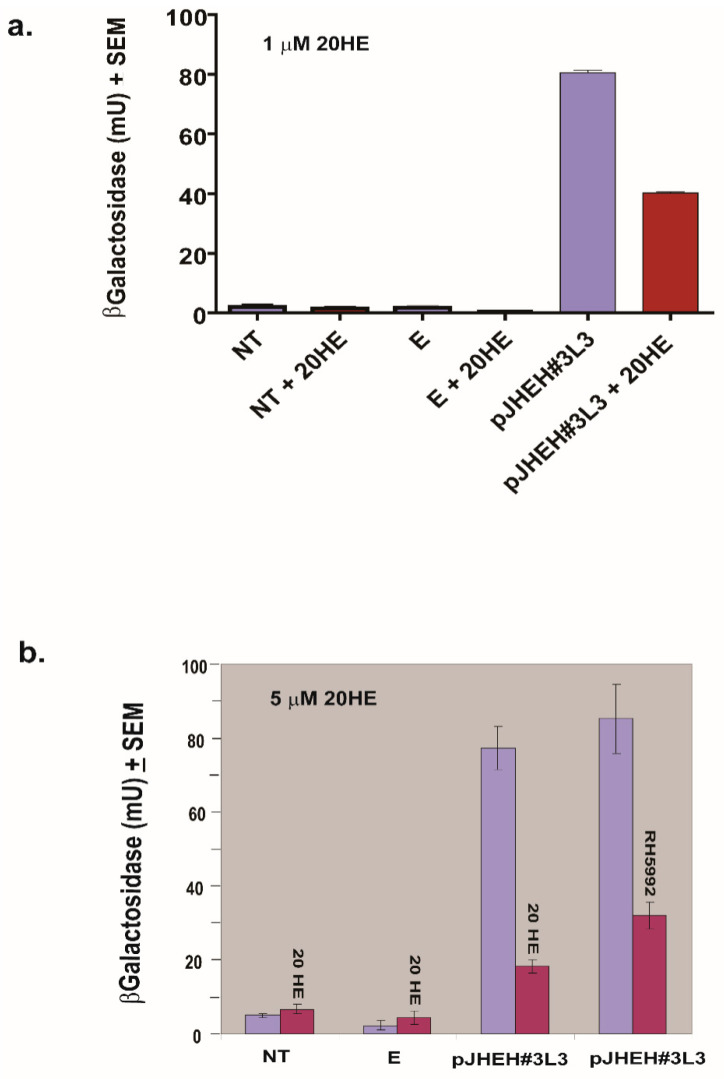
Effects of 20HE and its mimic (RH5992), 5 μM each, on the transcriptional activity of D.Mel2 transfected with *jheh* 3 promoter pJHEH#3L3. (**a**) Red colored bars show incubations with 20HE (1 μM), magenta-colored bars show incubations without 20HE. (**b**). Red-colored bars show incubations with 20HE or RH5992 (5 μM), magenta-colored bars show incubations without 20HE. NT—Non-transfected cells, E—cells transfected with an empty plasmid. See Figure 7 for promoter pJHEH#3L3 sequence length.

**Figure 12 biomolecules-12-00991-f012:**
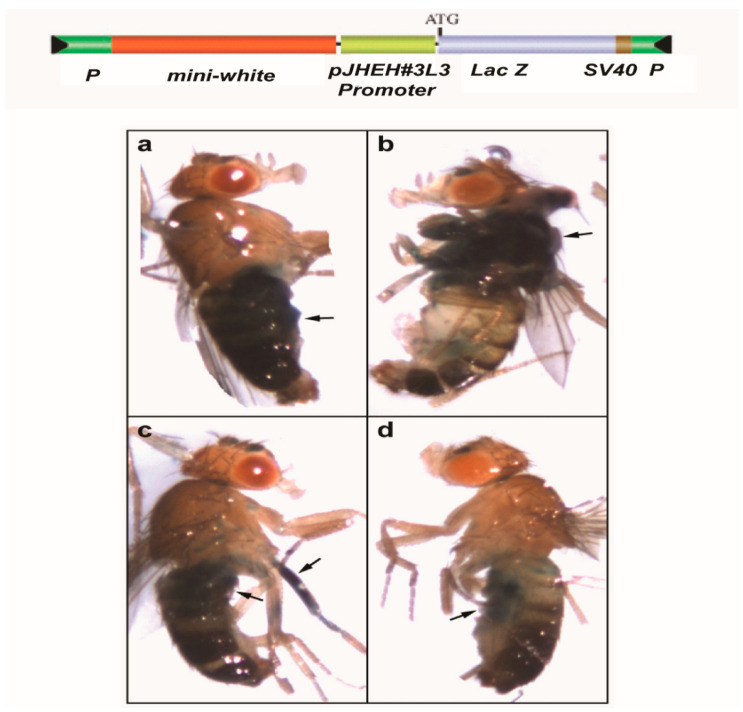
*D. melanogaster* flies that were transformed with *jheh* 3 promoter pJHEH#3L3 (colored green) in pCaSpeR-AUG-βgal behind *lac*Z (colored magenta) upper bar. Arrows point to β-galactosidase activity in the abdomen (**a**), in the thorax (**b**), in the leg muscle and upper abdomen next to the thorax (**c**) and the ventral part of the abdomen (**d**).

**Figure 13 biomolecules-12-00991-f013:**
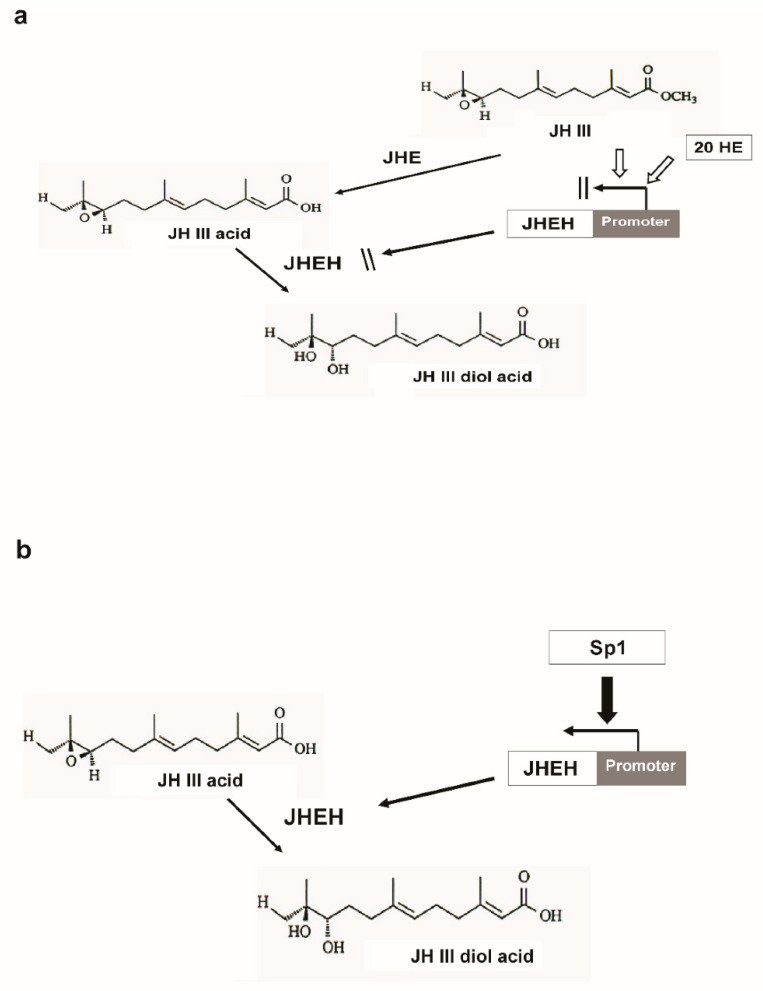
Degradative steps in JH III pathway. (**a**) JHEH promoter is inhibited by high titers of JH III and 20HE. During this time, JHEH is not synthesized. (**b**) After JH III is converted into JH IIIA (JH III acid) by JHE, the JHEH promoter is upregulated by TF Sp1, allowing the synthesis of JHEH, which converts JH IIIA into JH IIIAD (JH III acid diol).

**Table 1 biomolecules-12-00991-t001:** Geometric and thermodynamic quality of the three-dimensional models built for JHEH 1, 2 and 3 homodimers (A + B chain).

Calculated Parameters	JHEH 1	JHEH 2	JHEH 3
Ramachandran outliers	L94 (A)	M240 (A)	T58 (A)
	A72 (B)	G273 (A)	D66 (B)
QMEAN value	0.72	0.74	0.70
ANOLEA calculated of amino	28 over 450 (A)	26 over 439 (A)	16 over 437 (A)
acids residues exhibiting energetic	29 over 450 (B)	23 over 439 (B)	10 over 437 (B)
values over the threshold			

**Table 2 biomolecules-12-00991-t002:** In vivo metabolism of [12-^3^H]JH IIIA in female *D. melanogaster*.

Hours after Adult	Groups	JH IIIA	JH IIIAD
Eclosion	N	Cpm ± SEM	Cpm ± SEM
3	3	50,000 ± 6000	22,000 ± 2500
24	3	40,000 ± 5000	32,000 ± 1800
48	3	30,000 ± 6000	38,000 ± 4200
72	3	34,000 ± 4450	40,000 ± 6000

Groups of female *D. melanogaster* (20 per group) were treated with [12-^3^H]JH IIIA (0.1 μCi) per female at different times after adult eclosion and analyzed for JH IIIA and JH IIIAD 1 h after treatment using reversed-phase C_18_ HPLC [6]. Results are means of 3 determinations ± SEM.

## Data Availability

All the data reported in this manuscript is found in the manuscript including GenBank accession number and *D. melanogaster* genome accession numbers.

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
