# Peer review of "Cloning and Characterization of Drosophila melanogaster Juvenile Hormone Epoxide Hydrolases (JHEH) and Their Promoters"

_biomolecules, 2022, doi:10.3390/biom12070991_

Round 1
Reviewer 1 Report
The authors investigated the juvenile hormone epoxide hydrolases (JHEH) of Drosophila melanogaster as well as the promoter region of these genes. The subject will be of interest to people in the field, and should be published at some point. however, several points need to be addressed first.
1. The authors cloned the 3 JHEH from D. melanogaster. however, they never expressed them nor test their activity against any JHs or their metabolites. how can they be sure they are really JHEH? In 2003 by Taniai et al tried to clone the JHEH from D. melanogaster, but showed it had no activity against JH-III. It could be the same here. Until they showed it has activity against JH or JHA, they cannot call the clone proteins JHEH for sure.
Personally, i believe it will be simpler to just concentrate the manuscript (and title) on the promoter regions of 3 potential JHEH genes in D. melanogaster as most of the results are on that, and that the authors did nothing with the cloned proteins (the 3D models are nice looking but do not provide true new information into the proteins' activity and function).
2. In the abstract, the authors claimed that JHA is the preferred substrate for JHEH. There is no prove of that in this manuscript.
3. There are too many similar abbreviation. For the gene/protein, the authors need to use Arabic's numbers not Roman's, which are used for the JHs. At first read, one can think that JHEH-III hydrolyzes JH-III. But we do not know, so the confusion.
4. The second part of the introduction is weak. Until line 50, it reads well, then the authors tried to explain why the promoter region needs to be investigated. While not truly wrong, this part is confused and should be rewritten to guide the readers towards what is investigated here and its importance.
5. There is too much results presented in the methodology section. i counted 8 figures and 7 tables. Also the text goes beyond just describing the method. This section needs to be rewritten. Table/figure linked to method (e.g. primers) could be put in supplemental information.
6. There is far too much figures (14) and tables (7). some could be put in supplemental. Also, as explain above, just concentrating the results on the promoter experiments will help focus on the main point of the paper: promoter regulation of potential JHEH gene. The full characterization of the JHEH proteins could be a separate paper.
7. there is the problem with the formatting of the references.
8. between line 758 and 764, the author put a figure, and it cut the sentence.
Author Response
Please find our response in the attached file

Reviewer 2 Report
The manuscript entitled “Cloning and Characterization of Drosophila melanogaster Juvenile Hormone Epoxide Hydrolases (JHEH) and their promoters” describes sequentially the identification, cloning, and sequencing of the cDNAs of JHEH I, II, and III from female D. melanogaster, in a quest to study the role of JHEH in flies. The authors have analyzed the expression of JHEH in different fly tissues, viz., head thorax, gut, ovary, fat bodies, and the whole insect RNA extracts. Molecular 3D modeling further shows that the enzyme is a homodimer and binds juvenile hormone III acid (JH IIIA) at the catalytic groove in a way better than JH III, and that the JH IIIA could probably serve as a better substrate. Further, in order to quantify the strength of each JHEH promoter, the authors have utilized the b-galactosidase assay. The study exhibited the fact that the full-length JHEH I promoter showed the highest transcriptional activity compared to its shorter counterparts. On the contrary, JHEH II promoter activity showed that shorter promoter length increased the promoter’s transcriptional activity. Further, the authors have demonstrated an increase in β-galactosidase activity expressed by promoter pJHEH#3L3 which was found to be significantly higher than the longer and shorter segments of the JHEH I and II promoter sequences. To understand this discrepancy, the authors did an in-depth analysis of JHEH III promoter sequence and narrowed down their study revealing that TF Sp1 critically upregulates the JHEH III promoter. Taking the study a notch up the authors have also tried to study the effects of JH III, JH IIIA, 20Hydroxyecdysone (20HE), and farnesoic acid on the jheh III promoter.
Of note, this paper is a nice demonstration showing that JH IIIA is the preferred substrate of JHEH III, and jheh promoter is controlled by JH III and activated by TF Sp1.
The results support the findings, though there are a few points that should be considered before publication.
Though the manuscript is written with the utmost care, there are some typographical errors that the authors should take care of. For example, in Line 567, “ctin” should be corrected to “actin”.
Also, in line 235, D. melanogaster should be italicized.
Author Response
The two minor correction that were suggested by Referee 2 were done. ctin was converted in actin see line 522 in the revised manuscript. D. melanogaster is now italicized see line 220 in the revised manuscript. We would like to thank this referee for his suggestion and appreciate his comments.
Round 2
Reviewer 1 Report
thank you for the changes